# Real-Time Kinematically Synchronous Planning for Cooperative Manipulation of Multi-Arms Robot Using the Self-Organizing Competitive Neural Network

**DOI:** 10.3390/s23115120

**Published:** 2023-05-27

**Authors:** Hui Zhang, Hongzhe Jin, Mingda Ge, Jie Zhao

**Affiliations:** 1Institute of Robotics, Henan University of Technology, Zhengzhou 450001, China; huizh2021@haut.edu.cn; 2State Key Laboratory of Robotics and System, Harbin Institute of Technology, Harbin 150001, China; 19b908045@stu.hit.edu.cn (M.G.); jzhao@hit.edu.cn (J.Z.)

**Keywords:** multi-arms robot, collaborative manipulation, self-organizing competitive neural network, inner star rule, synchronous planning

## Abstract

This paper presents a real-time kinematically synchronous planning method for the collaborative manipulation of a multi-arms robot with physical coupling based on the self-organizing competitive neural network. This method defines the sub-bases for the configuration of multi-arms to obtain the Jacobian matrix of common degrees of freedom so that the sub-base motion converges along the direction for the total pose error of the end-effectors (EEs). Such a consideration ensures the uniformity of the EE motion before the error converges completely and contributes to the collaborative manipulation of multi-arms. An unsupervised competitive neural network model is raised to adaptively increase the convergence ratio of multi-arms via the online learning of the rules of the inner star. Then, combining with the defined sub-bases, the synchronous planning method is established to achieve the synchronous movement of multi-arms robot rapidly for collaborative manipulation. Theory analysis proves the stability of the multi-arms system via the Lyapunov theory. Various simulations and experiments demonstrate that the proposed kinematically synchronous planning method is feasible and applicable to different symmetric and asymmetric cooperative manipulation tasks for a multi-arms system.

## 1. Introduction

The development of artificial intelligence technology has facilitated the research on the autonomous manipulation of robot manipulators. Meanwhile, the increase in requirement of using robots to replace human hands’ manipulation has made the cooperative motion of the robot important in the autonomous operations (e.g., manipulating the rudder, using pliers or a wrench, carrying large objects, or other similar manual tasks in daily life) [1]. Various multi-arm robots, such as Baxter [2], YUMI [3], Justin [4], and Robonaut [5], have been proposed to satisfy the requirement because of their outstanding capability of cooperative manipulation in replacing humans.

Multi-arm robots can not only complete the manipulation task of single-arm robot (or robot manipulator) but also accomplish more complex cooperative manipulation task, which is attributed to their large workspace, more degrees of freedom (DOFs), and greater flexibility. The cooperative manipulation of robots has been studied a lot. The methods for the cooperative manipulation could be divided into force-based [6,7,8,9] and kinematics-based strategies [10,11,12,13,14,15,16,17,18,19,20,21,22,23,24,25,26,27,28,29]. 

The force-based strategies include two types: One relies on dynamics control, but the highly complicated nonlinear dynamic model makes it difficult to apply in robot system. The other one depends on position compensation control that adjusts the position of the end-effector (EE) to maintain a certain interaction force and that has been studied extensively. The hybrid control of force and position [6,7] and the impedance control [8,9] are the typical method using position compensation control. The hybrid control of force and position needs to accurately decompose the coordinate space of the EE into the position space and force space. The impedance control changes the contact force and the EE position according to the impedance value, but the impedance parameters must be adjusted adaptively to ensure the compliance of interaction in the cooperative manipulation. Although the variable impedance is proposed to improve the flexibility of interaction, the real impedance parameters are difficult to obtain. The intelligent control methods are proposed to simplify the control schemes and reduce the difficulty of modeling the robot control system caused by the strong coupling, time-varying, and uncertainty, such as a neural network [10]. In addition, the force-based strategies rely on force/torque sensors and the corresponding control algorithms and are mainly used in the non-redundant robot system [11].

The kinematics-based strategies are simple and easy to combine with an intelligent algorithm to realize robot autonomous manipulation, thereby resulting in the kinematics-based strategies becoming a research focus of the cooperative manipulation. Thus, the kinematics-based cooperative manipulation is mainly considered in this paper. Usually, the higher requirement for the motion synchronization among the arms needs to be satisfied in the kinematics-based cooperative manipulation. The existing kinematics-based strategies for the cooperative manipulation rely on motion planning, including leader–follower [12,13], cooperative-task space (CTS) [14,15,16,17,18,19], task-oriented [20,21], and intelligent approaches [11,22,23].

The leader–follower approach defines the leader arm and the follower arm for the robot system, and the follower carries out motion planning according to the movement of the leader [12,13]. In the CTS approach, robot arms without a leader-follower relationship are shared equally to achieve symmetrical cooperative manipulation tasks by defining the relative and absolute motions. The extended CTS approach was further proposed to accomplish asymmetric behavior and uncoordinated tasks [18,19]. The task-equation-based approach uses the general formula of the defined cooperative task to transform the coordinate system kinematics among arms to plan the motion of single arm [20,21]. The intelligent approach considers the collaborative task of arms as the constrained quadratic programming problems and utilizes the advanced neural network as the solver of the problems to control the arms motion [11,22]. The intelligent approaches could simplify the solving process of inverse kinematics for dual arms, but the adjustment of the EE attitudes is neglected in cooperative tasks. In Ref. [23], the dual-arm path-planning problem was transformed into a multi-objective optimization problem, and a co-evolutionary method with shared workspace was proposed to serve as a solver. A motion planner based on the kinematic model of a dual-arm robot system in [24] was designed to ensure grasping stability and dexterity. The movement under the relative motion frame of EEs was studied in [25] for the problem of two-hand assembly. An asymmetric task-planning method based on the Lyapunov theory was proposed in [26] to solve the problems of designing the control law of absolute motion tasks and updating the distribution of relative tasks among arms. Fractional-order derivative and the uncertain fractional-order differential equations were utilized to predict and correct motion trends, and the rationality of the method is verified by different cases [27]. A state feedback robust controller based on local information was designed to ensure that the states of multiple robots converge to a common motion state [28]. An extended Kalman filter collaborative algorithm based on the error compensation was proposed to reduce the state estimation error of delayed filtering in multi robot systems [29]. 

The key focus of the kinematics-based strategies planning is ensuring the motion synchronization in collaborative tasks. The motion of one arm is always taken as a reference to plan the motion of other arms in the existing literature. Moreover, the application of the existing approaches is more suitable for the multi-arms robot than those coupled with the common and fixed base. Even if there is an application in the robot with a dynamic coupling base, the dynamic coupling base is set to be stationary to ensure the cooperation between arms [24], and this is caused by the uncoordinated movement between the EEs of redundant arms. Few studies have been devoted to the kinematics-based planning of the redundant multi-arms with a dynamic coupling base to ensure the cooperation between the arms.

Therefore, on this basis of the previous studies [24,30,31], this paper proposed a novel kinematics-based synchronous planning for collaborative manipulation of the redundant multi-arms with dynamic coupling, which involves the inverse kinematics based on the sub-base method and the self-organizing competitive neural network. The following aspects differ from those in the existing literature.

A class of cooperative manipulation tasks of multi-arm robot described by generalized coordinate transformation matrix are summarized, such as carrying, manipulating the rudder, using a wrench, manipulating pliers, multi-station manipulation, and other similar cooperative manipulation. The configuration branch division of the multi-arm robot based on the sub-base method is proposed to identify each branch of the robot, and the inverse kinematics is calculated based on the damped least square method. Then, the multi-arms robot system can synchronously converge along the reducing direction of the total error. The self-organizing competitive neural network is proposed to promote motion synchronization between multiple arms, and it regards the cooperative movement between arms as the competitive relationship of neurons instead of relying on a defined arm motion as a reference in existing research. The inner star learning rule is used to change the neuron weight, and all neuron weight values are updated to adjust the motion of multi-arms in every instance of competitive learning. Thus, the multi-arm robot motion planning method is formed and realizes the synchronization of the arms’ motion state. The stability of the motion-planning algorithm is analyzed by using the Lyapunov theory and the inner star learning rule principle. The feasibility of the proposed method, the synchronization of motion state, and its applicability are demonstrated by dual-arm and three-arm robots with a dynamic base in different symmetric and asymmetric cooperative manipulations.

The remainder of this paper is organized as follows. Section 2 presents a type of cooperative manipulation, the sub-base description, and the Jacobian matrix definition for the multi-arms with physical coupling. Section 3 discusses the real-time kinematically synchronous planning method of collaborative manipulation based on the self-organizing competitive neural network and the stability. Section 4 and Section 5 provide the simulation and experimental results in different cases, respectively. Section 6 presents the conclusion. 

## 2. Cooperative Manipulation of Multi-Arms

### 2.1. A Type of Kinematically Cooperative Manipulation

In Figure 1, a type of cooperative manipulation for multi-arms is considered for this paper, such as carrying, manipulating rudder, using pliers and multi-station manipulation, etc. Figure 2 presents the common features for the kinematically cooperative manipulations that can be concluded as follows: 

(1) The coordinate system {T*_i_*} in or out of the specified object is referenced for the object pose, **t***_i_*, of each arm EE in real time. The object pose, **t***_i_*, is defined as follows:(1){Qi=QTi−RtiTiDi Rti(f^ti,ψti)=RTiO0RtiTiti=(Qi,f^ti·ψti)T,
where $Q_{i}$ is the desired position vector of the *i*-th EE in coordinate system {O_0_}. $Q_{T_{i}}$ is the vector from the coordinate system {O_0_} to the *i*-th coordinate system {T*_i_*}. $D_{i}$ is the position vector in coordinate system {O_0_} for **t***_i_*. RtiTi denotes the rotation matrix from the coordinate system for **t***_i_* to {T*_i_*}. RTiO0 refers to the rotation matrix from {T*_i_*} to {O_0_}. $R_{t_{i}}\left({\hat{f}}_{t_{i}},\psi_{t_{i}}\right)$ defines the rotational operator about the axis direction ${\hat{f}}_{t_{i}}$ by $\psi_{t_{i}}$ radians. **t***_i_*$\in R^{b\times 1}$. *b* = 3 for planar robot. *b* = 6 for spatial robot. *i* = 1, 2, …, N. N is the number of arms (or EEs).

(2) When the arms perform collaborative operations, the arms form a closed loop, and there is a certain motion constraint relationship between the EEs. Thus, the motion states of EEs from “1, 2, …, *i*” to “1’, 2’, …, *i*’” are kinematically consistent and synchronous. The descriptions for movement states are mutual during the execution of these tasks, like the motion error and the motion rate for each EE. Such common motion states of cooperative manipulations are expressed as the following problem in Equation (2), and Equation (2) is used as the judgment criteria for coordinated synchronous motion and is proved in Section 3.2.
(2)$$\{\begin{array}{c} \underset{t\to \infty }{\lim}e_{i}=t_{i}-s_{i}=0\;\;\;\;\;\;\left(a\right)\\ \underset{t\to \infty }{\lim}\left(\left\|v_{i}-{\dot{t}}_{i}\right\|\right)=0\;\;\;\;\;\;\left(b\right) \end{array},\;$$
where $e_{i}$ denotes the pose error of the *i*-th EE, $v_{i}$ defines the *i*-th EE velocity, $s_{i}$ is the pose of the *i*-th EE, and *t* is the time. Equation (2a) refers to the pose errors of EEs along the direction of error convergence, which not only can make the multi-arms reach the execution position of the manipulated object at the same time but also can ensure the minimization of the movement error between the arms in the process of cooperative manipulation. Equation (2b) guarantees that the movement speeds of the EEs are synchronous and that the manipulation speed of the EE is equal/close to the set or constraint value in the manipulation. Finally, the synchronization of arms during cooperative manipulation can be achieved. 

The *i*-th EE pose, velocity, and object pose are $s_{i}$, $v_{i}$ and $t_{i}$, respectively. $s_{i}$, $v_{i}$, $t_{i}\in R^{b\times 1}$. The *i*-th EE pose error and velocity can be calculated as follows: (3)$$e_{i}\left(T\right)=t_{i}\left(T\right)-s_{i}\left(T\right)\;,\;$$
(4)$$v_{i}\left(T\right)={\dot{s}}_{i}=\frac{s_{i}\left(T\right)-s_{i}\left(T-\Delta T\right)}{\Delta T},\;$$
where T signifies the current time, (T−ΔT) is the last sampling time, ΔT denotes the sampling period, s = (**s**_1_, **s**_2_, …, **s***_k_*)^T^, and **t** = (**t**_1_, **t**_2_, …, **t***_k_*)^T^.

### 2.2. Multi-Arms Robot with Physical Coupling

This paper considers the general configuration of the multi-arm robot with physical coupling to achieve the cooperative manipulation, as shown in Figure 3. Suppose that there are *r* joints of the multi-arms and each value of *θ_j_* is the joint angle. The completely joint configuration of the multi-arms is defined as **Θ** = (*θ*_1_, …, *θ_r_*)^T^, $\Theta \in R^{r\times 1}$. The pose mapping of the multi-arm robot from joint space to Cartesian space can be expressed as follows: (5)s=f(Θ),
and $s_{i}=f_{i}\left(\Theta \right)$ for the EE. 

The corresponding inverse mapping is as follows:(6)$$\Theta =f^{-1}\left(s\right).\;\;\;$$
where *f* is a highly nonlinear operator and difficult to solve. The iterative method via Jacobian matrix is used to approach the good solution of the mapping problem.

The traditional Jacobian matrix, **J**, is the partial derivative matrix of the whole chain system relative to the EE s. The Jacobian matrix is obtained via linear approximations of inverse kinematic problems. They linearly simulate the motion of the EE with respect to the instantaneous system changes of the link translation and joint angle. The traditional Jacobian matrix, **J**, is a function of the joint angle, Θ, defined as follows: (7)$$J{\left(\Theta \right)}_{ij}={\left(\frac{\partial s_{i}}{\partial \theta_{j}}\right)}_{ij},$$
where *I* = 1, …, N. *j* = 1, …, *r*. $J{\left(\Theta \right)}_{ij}\in R^{b\times 1}$.

### 2.3. Definition of Sub-Bases 

The traditional Jacobian matrix can ensure that each EE converges along its own error reduction direction. Unlike the traditional Jacobian matrix, J, the sub-bases are defined to make the EEs converge along the reducing direction of the system’s total error and guarantee that the EEs converge at the same time.

This paper defines that the nodes with multiple branches as the sub-bases for the multi-arms configuration, as shown in Figure 3. The Jacobian matrix is modified as follows:(8)$$J\left(\Theta \right)=\mathrm{diag}\left(J_{1,1},J_{n,1},\ldots ,J_{n,k},J_{1},\ldots ,J_{N}\right).$$

For the 1-th sub-base pose, $P_{1,1}$, the corresponding element of the Jacobian matrix is as follows:(9)$$J_{1,1}{\left(\Theta \right)}_{j}=\overset{N}{\underset{i=1}{\sum }}{\left(\frac{\partial s_{i}}{\partial \theta_{j}}\right)}_{j},$$
where $J_{1,1}{\left(\Theta \right)}_{j}\in R^{b\times 1}$. $J_{1,1}\left(\Theta \right)\in R^{b\times M_{0,1}}$. $\theta_{j}$ belongs to the chain $P_{0}—P_{1,1}$ with M_0,1_ DoFs.

For the *n*,*k*-th sub-base pose, $P_{n,k}$, the corresponding element of the Jacobian matrix is as follows:(10)$$J_{n,k}{\left(\Theta \right)}_{j}={\left[\overset{N}{\underset{i=N-N_{k}+1}{\sum }}\left(\frac{\partial s_{i}}{\partial \theta_{j}}+\ldots +\frac{\partial s_{N}}{\partial \theta_{j}}\right)\right]}_{j},$$
where $J_{n,k}{\left(\Theta \right)}_{j}\in R^{b\times 1}$. $J_{n,k}\left(\Theta \right)\in R^{b\times M_{n,k}}$. $\theta_{j}$ belongs to the chain $P_{1,1}—P_{n,k}$ with M*_n_*_,*k*_ DoFs. By analogy, the Jacobian matrix corresponding to other sub-bases can be obtained.

For $J_{1},\ldots ,J_{N}$ without common degrees of freedom, the corresponding elements can be obtained according to (6).
(11)$$N=N_{1}+N_{2}+\ldots +N_{k},\;$$
(12)$$r=M_{0,1}+M_{n,1}+\ldots +M_{n,k}+M_{1}+\ldots +M_{N}.\;$$

The velocities of 1-th sub-base and *n*,*k*-th sub-base are calculated by the following:(13)$${\dot{P}}_{1,1}=\eta_{1}\cdot \frac{1}{N}\cdot \overset{N}{\underset{i=1}{\sum }}k_{p}\cdot \frac{d\left(t_{i}-s_{i}\right)}{dt},$$
(14)$${\dot{P}}_{n,k}=\eta_{n,k}\cdot \frac{1}{N_{k}}\cdot \overset{N}{\underset{i=N-N_{k}+1}{\sum }}k_{p}\cdot \frac{d\left(t_{i}-s_{i}\right)}{dt},$$
where $\eta_{1}$ and $\eta_{n,k}$ are the gain coefficient. N and N_k_ are fixed value and related to the configuration of multi-arms.

The inverse kinematics is as follows:(15)$$\dot{\Theta }=J^{*}\left(\Theta \right)\dot{S}=J^{*}\left(\Theta \right){\left({\dot{P}}_{1,1}\;\;\;{\dot{P}}_{n,1}\;\;\;\cdots \;\;\;{\dot{P}}_{n,k}\;\;\;\dot{s}\right)}^{T},\;$$
where **J**^*^ denotes the pseudo-inverse of Jacobian matrix, **J**(**Θ**), based on the damped least squares method, and **J***^*^*= **J**^T^(**JJ**^T^ + *λ***I**)^−1^. $J\left(\Theta \right)\in R^{b\left(k+N+1\right)\times r}$. $J^{*}\in R^{r\times b\left(k+N+1\right)}$. *λ* (*λ* > 0) represents the damping factor that can handle the ill-conditioned **J** in the neighborhood of singular configurations for redundant manipulators and guarantee the EEs with the minimum possible deviation at all configurations. **I** is a unit matrix with the dimension b(k+N+1) × b(k+N+1). In accordance with the traditional fixed proportion-based method [31] for the real-time tracking of a given object pose, $\dot{t}$, the EE velocities are planned as follows:(16)$$\dot{s}=\dot{t}+k_{p}\cdot \left(t-s\right),$$
where $k_{p}$ is the gain coefficient.

### 2.4. Iteration for Multi-Arms Robot Motion

The iterative method is utilized to achieve the real-time movement of multi-arms via updating the joint angles, **Θ,** according to (17).
(17)Θ(T)=Θ(T−ΔT)+ΔΘ. 
where ΔΘ deduced from (15) becomes
(18)$$\Delta \Theta \approx J^{*}\left(\Theta \right)\Delta S=J^{*}\left(\Theta \right){\left(\Delta P_{1,1}\;\;\;\Delta P_{n,1}\;\;\;\cdots \;\;\;\Delta P_{n,k}\;\;\;\Delta s\right)}^{T}\;\;=J^{*}\left(\Theta \right)\left[\begin{array}{c} \eta_{1}\cdot \frac{1}{N}\cdot \overset{N}{\underset{i=1}{\sum }}\mu \cdot \left(t_{i}-s_{i}\right)\\ \eta_{n,1}\cdot \frac{1}{N_{1}}\cdot \overset{N_{1}}{\underset{i=1}{\sum }}\mu \cdot \left(t_{i}-s_{i}\right)\\ \vdots \\ \eta_{n,k}\cdot \frac{1}{N_{k}}\cdot \overset{N}{\underset{i=N-N_{k}+1}{\sum }}\mu \cdot \left(t_{i}-s_{i}\right)\\ \Delta t+\mu \cdot \left(t-s\right) \end{array}\right],$$
where *μ* = *k_p_*·ΔT, and *μ* < 1. Moreover, Δt represents the changing pose of the object at a sampling time interval, ΔT.

Then, according to the sub-base method, the movement of the multi-arm robot can be achieved. The sub-base motion facilitates the synchronous convergence, and the synchronous performance is more obvious when the common DoFs are enough. The corresponding verifications are presented in Section 4 and Section 5.2.

## 3. Kinematically Synchronous Planning

### 3.1. Synchronous Planning Using Self-Organizing Competitive Neural Network

The DoFs of the sub-base are not always enough to ensure the system convergence along the reducing error direction for the pose of the EEs, thereby resulting in the asynchronous EE motion. Thus, the self-organizing competitive neural network based on the rule of inner star model is proposed to adjust the synchronism of multi-arms movement. Equation (2) indicates that the error $e_{i}$ and the $v_{i}$ will tend to a stable value. In accordance with the principle of the self-organizing competitive neural network, the kinematically synchronous planning for the multi-arm robot is designed as shown in Figure 4.

The learning rule of the inner star model defines the weight updating as follows:(19)$$\Delta w_{i}=\eta \left(P_{i}-w_{i}\right)Y_{i},$$
where η denotes the learning rate; $P_{i}$ is the *i*-th input element of the neuron and the minimum pose velocity error norm, min($\left\|v_{i}\left(T\right)-{\dot{t}}_{i}\right\|$); $w_{i}$ is the weight value; *i* = 1, 2, …, N; N is the number of arms (or EE); and $Y_{i}$ is the value of output neuron and is defined as
(20)$$Y_{i}=\{\begin{array}{c} 1,\;\;\;\;\;\;\mathrm{if}\;P_{i}^{\prime }P_{i}>\varepsilon \\ \;0,\;\;\;\;\;\;\mathrm{Otherwise}\; \end{array},$$
where
(21)$$\varepsilon =\frac{PP}{N}.\;$$

The input element, $P_{i}$, and the weight value, $w_{i}$, are defined as
(22)$$P_{i}=\|\tilde{v}\|=\min\left(\left\|v_{1}\left(T\right)-{\dot{t}}_{1}\|,\|v_{2}\left(T\right)-{\dot{t}}_{2}\|,\ldots ,\|v_{N}\left(T\right)-{\dot{t}}_{N}\right\|\right),$$
(23)$$w_{i}=\|v_{i}\left(T\right)-{\dot{t}}_{i}\|.$$

The input vector P is constituted by $P_{i}$ and defined as
(24)$$P={\left(P_{1},\;P_{2},\ldots ,P_{N}\right)}^{T},\;P\in R^{N\times 1}.\;\;\;\;$$

Since the minimum of $\left\|\tilde{v}\right\|$ is used as an input and each EE may become the one with the minimum of $\left\|\tilde{v}\right\|$, the proposed method makes the planner no longer use a manipulator as a reference as in the existing literature. In each period, the new input vectors, $P^{\prime }$**,** in neural networks are defined as
(25)$$P^{\prime }={\left(P_{1}^{\prime },P_{2}^{\prime },\ldots ,P_{N}^{\prime }\right)}^{T}.\;$$
where
(26)$$P_{i}^{\prime }=\|v_{i}\left(T\right)-{\dot{t}}_{i}\|.\;$$

In order to make the system quickly reach the state of synchronization, the winning weight value will get more rewards in a cycle. The new weight value, $\Delta {\hat{w}}_{i}$, is designed as
(27)$$\Delta {\hat{w}}_{i}=\Delta w_{i}\cdot \tanh\left(\frac{c_{i}\|v_{i}\left(T\right)-{\dot{t}}_{i}\|}{\sum_{i=1}^{N}\|v_{i}\left(T\right)-{\dot{t}}_{i}\|+\sigma_{i}}\right),\;$$
where $\sigma_{i}$ is a small positive real number; and $c_{i}$ denotes a positive real number to adjust the updating slope of neuron weight and contributes to the synchronization of the EE motion. The response time of converges is shorter when the parameter $c_{i}$ becomes larger. 

The *i*-th EE’s planned related velocity is deduced according to the iterative method as follows:(28)$${\hat{v}}_{i}\left(T\right)=\mu \cdot \left(t_{i}-s_{i}\right)\cdot \left[1-\frac{\Delta {\hat{w}}_{i}}{\|v_{i}\left(T\right)-{\dot{t}}_{i}\|+\delta_{i}}\right].\;$$

Then, the planned velocity for EEs is
(29)$${\bar{v}}_{i}\left(T\right)={\hat{v}}_{i}\left(T\right)+\Delta t_{i}.\;$$

Thus, (18) becomes
(30)$$\Delta \Theta \approx J^{*}\left(\Theta \right)\Delta S=J^{*}\left(\Theta \right)\left[\begin{array}{c} \eta_{1}\cdot \frac{1}{N}\cdot \overset{N}{\underset{i=1}{\sum }}{\hat{v}}_{i}\left(T\right)\\ \eta_{n,1}\cdot \frac{1}{N_{1}}\cdot \overset{N_{1}}{\underset{i=1}{\sum }}{\hat{v}}_{i}\left(T\right)\\ \vdots \\ \;\eta_{n,k}\cdot \frac{1}{N_{k}}\cdot \overset{N}{\underset{i=N-N_{k}+1}{\sum }}{\hat{v}}_{i}\left(T\right)\\ \bar{v}\left(T\right) \end{array}\right],$$
where $\bar{v}\left(T\right)={\left[{\bar{v}}_{1}\left(T\right),{\bar{v}}_{2}\left(T\right),\ldots ,{\bar{v}}_{N}\left(T\right)\right]}^{T}$. The learning rate *η*, *η*_1_ and *η*_n,k_ are related to the step size for each iteration. The response time of converges is shorter when the learning rate becomes larger. 

### 3.2. Stability Analysis

Supposing that the learning rate, i.e., η, $\eta_{1}$, $\eta_{n,1}$, …, $\eta_{n,k}$, is small, based on the rule of inner star model, the weight value with the minimum EE pose error, $e_{\min}$, is never updated, and the minimum pose error changes in a cycle is $\Delta e_{\min}\left(T\right)$. Thus, for the EE with the minimum pose error, the Lyapunov function is defined as follows: (31)$$V\left(T\right)=\frac{1}{2}e_{\min}^{2}\left(T\right),$$
(32)$$V\left(T+\Delta T\right)=\frac{1}{2}e_{\min}^{2}\left(T+\Delta T\right),\;$$

The change of the pose error is
(33)$$e_{\min}\left(T+\Delta T\right)=e_{\min}\left(T\right)-\Delta e_{\min}\left(T\right),\;$$

For the EE with the minimum pose error, $e_{\min}\left(T\right)$, the planned iterative step based on the fixed proportion method [31] is $\mu e_{\min}\left(T\right)$ in each control cycle, as shown in Figure 5. However, the actual motion of EE will have a small deviation, φ**,** as follows:(34)$$\Delta e_{\min}\left(T\right)=\mu e_{\min}\left(T\right)+\varphi ,$$
where 0 < *μ* < 1.

Due to the high-precision motion of the advanced manipulator and the small learning rate in the proposed method, the deviation, φ**,** is usually in a very small range. The norm of the deviation ‖φ‖ is less than $\left\|\mu e_{\min}\left(T\right)\right\|$ to ensure that the EE can move along the planned path, i.e., $\|\varphi \|<\|\mu e_{\min}\left(T)\right\|$. Therefore,
(35)$$\Delta V\left(T\right)=\frac{1}{2}e_{\min}^{2}\left(T+\Delta T\right)-\frac{1}{2}e_{\min}^{2}\left(T\right)\;=\frac{1}{2}\left\{{\left[e_{\min}\left(T\right)-\mu e_{\min}\left(T\right)-\varphi \right]}^{2}-e_{\min}^{2}\left(T\right)\right\}\;=\frac{1}{2}\left\{{\left[\left(1-\mu \right)e_{min}\left(T\right)-\varphi \right]}^{2}-e_{min}^{2}\left(T\right)\right\}\;\leq \frac{1}{2}\left\{{\left[\left(1-\mu \right)e_{min}\left(T\right)+\|\varphi \|\right]}^{2}-e_{min}^{2}\left(T\right)\right\}\;<\frac{1}{2}\left\{{\left[\left(1-\mu \right)e_{min}\left(T\right)+\mu e_{min}\left(T\right)\right]}^{2}-e_{min}^{2}\left(T\right)\right\}\;=\frac{1}{2}\left\{{\left[\left(1-\mu \right)e_{min}\left(T\right)+\mu e_{min}\left(T\right)\right]}^{2}-e_{min}{\left(T\right)}^{2}\right\}\;=0$$

Thus,
(36)$$\underset{T\to \infty }{\lim}e_{\min}\left(T\right)=0,\;$$
and the minimum pose error, $e_{\min}$, is convergent. 

Based on Equations (3) and (4), the derivation of ${\dot{e}}_{\min}\left(T\right)$ is
$${\dot{e}}_{\min}\left(T\right)={\dot{t}}_{\min}\left(T\right)-v_{\min}\left(T\right)={\dot{t}}_{\min}\left(T\right)-{\dot{s}}_{\min}\left(T\right)\;\;=\frac{t_{\min}\left(T\right)-t_{\min}\left(T-\Delta T\right)}{\Delta T}-\frac{s_{\min}\left(T\right)-s_{\min}\left(T-\Delta T\right)}{\Delta T}\;=\frac{t_{\min}\left(T\right)-s_{\min}\left(T\right)}{\Delta T}-\frac{t_{\min}\left(T-\Delta T\right)-s_{\min}\left(T-\Delta T\right)}{\Delta T}\;=\frac{1}{\Delta T}\left[e_{\min}\left(T\right)-e_{\min}\left(T-\Delta T\right)\right]\;=\frac{1}{\Delta T}\left[e_{\min}\left(T-\Delta T\right)\right]\;=\frac{1}{\Delta T}\left[\mu e_{\min}\left(T-\Delta T\right)+\varphi \right]$$

Since $\|\varphi \|<\|\mu e_{\min}\left(T)\right\|$ and $\underset{T\to \infty }{\lim}e_{\min}\left(T\right)=0$,
(37)$$\underset{T\to \infty }{\lim}{\dot{e}}_{\min}\left(T\right)=0,\;$$

According to the rule of inner star model, the input and the weight ultimately become equal, as follows:(38)$$P_{i}=w_{i}.$$

This is
(39)$$\|\tilde{v}\|=\|{\dot{e}}_{\min}\left(T\right)\|=\|v_{i}\left(T\right)-{\dot{t}}_{i}\|=0,$$
(40)$$\Delta {\hat{w}}_{i}=\Delta w_{i}=0.\;$$

Then,
(41)$$\underset{T\to \infty }{\lim}\|e_{i}\left(T\right)\|=\underset{T\to \infty }{\lim}\|e_{\min}\left(T\right)\|=0.$$
(42)$$\underset{T\to \infty }{\lim}\|{\dot{e}}_{i}\left(T\right)\|=\underset{T\to \infty }{\lim}\|{\dot{e}}_{\min}\left(T\right)\|=0.$$

Therefore, the proposed planning method based on the self-organizing competitive neural network is convergent, and Equation (2) is proved to be valid to achieve motion synchronization.

## 4. Simulation

The three-arm robot with 15-DoFs is utilized to provide the contrast simulations for inverse kinematics by using inverse kinematics based on the sub-bases and the traditional method [31], as shown in Figure 6. The common DoFs are 3 that are enough for all EEs to make the pose error converge along the decreasing direction of the total error simultaneously. The synchronization performances of EE movements are compared by tracking the static-designated location on the specified-carried object. The robot configuration parameters, initial parameters, and kinematics parameters are presented in Table 1, Table 2 and Table 3. The poses of the static object are **t**_1_ = (2.49 m, 2.79 m, 1.536 rad)^T^, **t**_2_ = (5.0 m, −0.70 m, 1.536 rad)^T^, and **t**_3_ = (4.49 m, 1.50 m, 1.536 rad)^T^. Moreover, **t** = (**t**_1_, **t**_2_, **t**_3_)^T^, and $\dot{t}$ = **0**. The simulation results are presented in Figure 7 and Figure 8. 

Both methods can make the EEs arrive at the designated location of the specified object, as illustrated in Figure 7a and Figure 8a. The joint angles are shown in Figure 7b and Figure 8b. Due to $\dot{t}=0$, $\Delta e_{i}\left(T\right)$ is equal to $v_{i}\left(T\right)\Delta T$, and the velocity curves of the EE pose are similar with those of the EE pose error, as shown in Figure 7c–f and Figure 8c–f. Because of the sub-base motion, the convergence of the EE pose error is much faster than that based on the traditional method. Furthermore, the EE pose velocities form uniform motion states before complete convergence in Figure 8c,d. Thus, the proposed sub-base method is conducive to the synchronization from the initial motion state to the cooperative motion state and improves the efficiency for the carrying of the multi-arm robot.

## 5. Experimental Verification

### 5.1. Experimental Setup

The two-arm robot with 13-DoFs is used to compare the proposed synchronous planning for the collaborative manipulation, and 1-DoF is common in the base joint, as shown in Figure 9. The D-H parameters of the two-arm robot are presented in Table 4. In Figure 10, the principle of the experimental setup can be briefly described as follows. 

(1)The global depth camera transfers the observed frame of depth and color images to the computer by using a universal serial bus (USB) in real time. The robot operating system (ROS) node runs in the computer and extracts the position information of the object from each frame image. The vision-processing procedures in the computer are developed based on the morphology, using the Open Source Computer Vision Library and the camera Software Development Kit.(2)The joint feedback data, **Θ**, of the robot are transmitted to the TMS320F28335 controller through the Controller Area Network (CAN) bus. At the same time, the controller transfers the received joint data, **Θ**, to the ROS nodes through the Serial Communication Interface (SCI) bus in real time.(3)The ROS nodes receive the data (**x***_obj_*, **Θ**) and execute the forward and inverse kinematics calculation and motion-planning algorithm. The real-time control command data, **Θ***_d_* and ${\dot{\Theta }}_{d}$, are sent to the robot joint actuator through the RS485 bus to control the motion. The baud rate of the CAN bus, USB serial bus, SCI bus, and RS485 bus is set to 1 MHz. The control period of the whole system containing the proposed kinematically synchronous planning method is less than 10 ms.

### 5.2. Synchronous Planning Experiments

(1)Collaboration Carrying

The collaboration carrying an object was provided to verify the feasibility of the proposed planning method, as illustrated in Figure 11. The parameters for carrying are shown in Table 5 and Table 6, respectively. The carried object has translational and rotational motion. The rotation center position is (−1.5 mm, −342.7 mm, −0.85 mm)^T^. The EEs of left and right arms reach the initial locations, as shown in Figure 11a. The tracked initial locations on the carried object are $t_{1}^{\mathrm{init}}\;$ = (115.841, −342.7, −0.85, 1.2092, −1.2092, −1.2092)^T^, $t_{2}^{\mathrm{init}}\;$ = (−118.841, −342.7, −0.85, 1.2092, −1.2092, −1.2092)^T^. At 6.75 s, two arms begin to carry the object, as shown in Figure 11b. The rotation angle relative to the rotation center is 0.0005 rad, and the translational motion of the rotation center is 0.15 mm in a cycle period, ΔT. During the collaboration process, the position of the tracked locations on the carried object moves back and forth in a 5 s cycle, and the tracked pose trajectory in the initial 2.25 s (from 6.75 s to 9 s) is as follows:(43)$$t_{1}=t_{1}^{\mathrm{init}}-\left(\begin{array}{c} 117.341\cdot \left[\sin\left(0.0005\left(T-6.75\right)+0.0005\Delta T\right)-\sin\left(0.0005\left(T-6.75\right)\right)\right]+0.15\left(T-6.75\right)\\ 117.341\cdot \left[\cos\left(0.0005\left(T-6.75\right)+0.0005\Delta T\right)-\cos\left(0.0005\left(T-6.75\right)\right)\right]\\ -0.15\left(T-6.75\right)\\ {\hat{f}}_{1}\cdot \psi_{1} \end{array}\right)$$
(44)$$t_{2}=t_{2}^{\mathrm{init}}+\left(\begin{array}{c} 117.341\cdot \left[\sin\left(0.0005\left(T-6.75\right)+0.0005\Delta T\right)-\sin\left(0.0005\left(T-6.75\right)\right)\right]+0.15\left(T-6.75\right)\\ 117.341\cdot \left[\cos\left(0.0005\left(T-6.75\right)+0.0005\Delta T\right)-\cos\left(0.0005\left(T-6.75\right)\right)\right]\\ 0.15\left(T-6.75\right)\\ {\hat{f}}_{2}\cdot \psi_{2} \end{array}\right)\;$$
where
${\hat{f}}_{1}$, φ_1_, ${\hat{f}}_{2}$, and φ_2_ can be obtained according to the following equations:(45)$$R_{1}({\hat{f}}_{1},\varphi_{1})=R_{2}({\hat{f}}_{2},\varphi_{2})\;=\left[\begin{array}{ccc} \cos\left(-\frac{\pi }{2}-0.0005\left(T-6.75\right)\right) & -\sin\left(-\frac{\pi }{2}-0.0005\left(T-6.75\right)\right) & 0\\ \sin\left(-\frac{\pi }{2}-0.0005\left(T-6.75\right)\right) & \cos\left(-\frac{\pi }{2}-0.0005\left(T-6.75\right)\right) & 0\\ 0 & 0 & 1 \end{array}\right]\left[\begin{array}{ccc} 1 & 0 & 0\\ 0 & 0 & 1\\ 0 & -1 & 0 \end{array}\right]\;$$

All the parameters in Table 5 and Table 6, the rotation center position, $t_{1}^{\mathrm{init}}$, and $t_{1}^{\mathrm{init}}$, were used in Figure 4. The inverse kinematics is based on Equation (30) to obtain the joint angle, **Θ**. The change of the new weight value, $\Delta {\hat{w}}_{i}$, of the self-organizing competitive neural network is obtained according to Equation (27). The learning rate, i.e., η, $\eta_{1}$, $\eta_{n,1}$, …, $\eta_{n,k}$, is used to calculate the carrying step size of EE motion in Cartesian space. With the continuous learning and competition of the proposed planning method, the position velocities of EEs gradually form a consistent movement. Figure 12a,b show the motion paths of EEs and joint trajectories, respectively. Figure 12c,d show the periodic change of pose in Cartesian space where the *Y*-axis position, the *Z*-axis position, and the attitude of the EEs remain the same. The position velocity and the attitude velocity reach synchronous motion states when the position velocity error and the attitude velocity error decrease to 0, as illustrated in Figure 12e–h.

(2)Manipulating the pliers

Figure 13 shows that the two arms manipulated a pair of pliers. The parameters are illustrated in Table 5 and Table 6. The rotation center position is (−50.0 mm, −347.55 mm, and 0.85 mm)^T^. The length of the pliers’ handle is 135 mm. The tracked initial locations on the carried object for the EEs of left and right arms are $t_{1}^{\mathrm{init}}$ = (86.841, −212.7, 0.85, 1.2092, −1.2092, −1.2092)^T^ and $t_{2}^{\mathrm{init}}\;$ = (−186.841, −212.7, 0.85, 1.2092, −1.2092, −1.2092)^T^. At 6.25 s, two arms begin to manipulate the pliers, as shown in Figure 14b. During the collaboration process, the position of the tracked locations on the pliers moves back and forth in a 3 s cycle, and the tracked pose trajectory in the initial 1.5 s (from 6.25 s to 7.75 s) is as follows:(46)$$t_{1}=t_{1}^{\mathrm{init}}+\left(\begin{array}{c} 135\cdot \left[\sin\left(0.0005\left(T-6.25\right)+0.0005\Delta T\right)-\sin\left(0.0005\left(T-6.25\right)\right)\right]\\ 135\cdot \left[\cos\left(0.0005\left(T-6.25\right)+0.0005\Delta T\right)-\cos\left(0.0005\left(T-6.25\right)\right)\right]\\ 0\\ {\hat{f}}_{1}\cdot \psi_{1} \end{array}\right)\;$$
(47)$$t_{2}=t_{2}^{\mathrm{init}}+\left(\begin{array}{c} 135\cdot \left[\sin\left(-0.0005\left(T-6.25\right)-0.0005\Delta T\right)-\sin\left(-0.0005\left(T-6.25\right)\right)\right]\\ 135\cdot \left[\cos\left(0.0005\left(T-6.25\right)+0.0005\Delta T\right)-\cos\left(0.0005\left(T-6.25\right)\right)\right]\\ 0\\ {\hat{f}}_{2}\cdot \psi_{2} \end{array}\right)$$
where ${\hat{f}}_{1}$, $\psi_{1}$, ${\hat{f}}_{2}$, and $\psi_{2}$ can be obtained according to the following equations:
(48)$$R_{1}({\hat{f}}_{1},\varphi_{1})=\left[\begin{array}{ccc} \cos\left(-\frac{\pi }{2}-0.0005\left(T-6.25\right)\right) & -\sin\left(-\frac{\pi }{2}-0.0005\left(T-6.25\right)\right) & 0\\ \sin\left(-\frac{\pi }{2}-0.0005\left(T-6.25\right)\right) & \cos\left(-\frac{\pi }{2}-0.0005\left(T-6.25\right)\right) & 0\\ 0 & 0 & 1 \end{array}\right]\left[\begin{array}{ccc} 1 & 0 & 0\\ 0 & 0 & 1\\ 0 & -1 & 0 \end{array}\right]\;$$
(49)$$R_{2}({\hat{f}}_{2},\varphi_{2})=\left[\begin{array}{ccc} \cos\left(-\frac{\pi }{2}+0.0005\left(T-6.25\right)\right) & -\sin\left(-\frac{\pi }{2}+0.0005\left(T-6.25\right)\right) & 0\\ \sin\left(-\frac{\pi }{2}+0.0005\left(T-6.25\right)\right) & \cos\left(-\frac{\pi }{2}+0.0005\left(T-6.25\right)\right) & 0\\ 0 & 0 & 1 \end{array}\right]\left[\begin{array}{ccc} 1 & 0 & 0\\ 0 & 0 & 1\\ 0 & -1 & 0 \end{array}\right]\;$$


The inverse kinematics is based on Equation (30) to obtain the joint angle, **Θ**. The change of the new weight value, $\cdot {\hat{w}}_{i}$, is obtained according to Equation (27) and used to adjust the EE motions according to Equation (28). The learning rate, i.e., η, $\eta_{1}$, $\eta_{n,1}$, …, $\eta_{n,k}$, is used to calculate the step size of manipulating the pilers in the Cartesian space. The proposed planning method makes the position velocities of EEs gradually form a consistent movement. Figure 14a,b show the motion paths of EEs and joint trajectories, respectively. Figure 14c,d show the periodic change of pose in Cartesian space, where the *Y*-axis position, the *Z*-axis position, and the attitude of the EEs remain the same. The position velocity and the attitude velocity of the EEs are almost the same and showed a small error when synchronous motion states were reached, as illustrated in Figure 14e–h. 

(3)Manipulating a rudder

Manipulating a rudder is illustrated in Figure 15. The parameters are shown in Table 5 and Table 6. The rotation center position is (0.0 mm, −342.7 mm, −0.85 mm)^T^. The diameter of the rudder is 136.84 mm. The tracked initial locations on the carried object for the EEs of left and right arms are $t_{1}^{\mathrm{init}}\;$ = (136.84, −342.7, −0.85, 1.2092, −1.2092, −1.2092)^T^ and $t_{2}^{\mathrm{init}}\;$ = (−136.84, −342.7, 0.85, 1.2092, −1.2092, −1.2092)^T^. At 6.75 s, two arms begin to manipulate the rudder, as shown in Figure 16b. During the collaboration process, the position of the tracked locations on the rudder moves back and forth in a 2.5 s cycle, and the tracked pose trajectory in the initial 1.25 s (from 6.75 s to 8 s) is as follows:(50)$$t_{1}=t_{1}^{\mathrm{init}}-\left(\begin{array}{c} 136.841\cdot \left[\sin\left(0.0005\left(T-6.75\right)+0.0005\Delta T\right)-\sin\left(0.0005\left(T-6.75\right)\right)\right]\\ 136.841\cdot \left[\cos\left(0.0005\left(T-6.75\right)+0.0005\Delta T\right)-\cos\left(0.0005\left(T-6.75\right)\right)\right]\\ 0\\ {\hat{f}}_{1}\cdot \psi_{1} \end{array}\right)$$
(51)$$t_{2}=t_{2}^{\mathrm{init}}+\left(\begin{array}{c} 136.841\cdot \left[\sin\left(0.0005\left(T-6.75\right)+0.0005\Delta T\right)-\sin\left(0.0005\left(T-6.75\right)\right)\right]\\ 136.841\cdot \left[\cos\left(0.0005\left(T-6.75\right)+0.0005\Delta T\right)-\cos\left(0.0005\left(T-6.75\right)\right)\right]\\ 0\\ {\hat{f}}_{2}\cdot \psi_{2} \end{array}\right)\;$$
where ${\hat{f}}_{1}$, $\psi_{1}$, ${\hat{f}}_{2}$, and $\psi_{2}$ can be obtained according to the following equations:(52)$$R_{1}({\hat{f}}_{1},\varphi_{1})=R_{2}({\hat{f}}_{2},\varphi_{2})\;=\left[\begin{array}{ccc} \cos\left(-\frac{\pi }{2}-0.0005\left(T-6.75\right)\right) & -\sin\left(-\frac{\pi }{2}-0.0005\left(T-6.75\right)\right) & 0\\ \sin\left(-\frac{\pi }{2}-0.0005\left(T-6.75\right)\right) & \cos\left(-\frac{\pi }{2}-0.0005\left(T-6.75\right)\right) & 0\\ 0 & 0 & 1 \end{array}\right]\left[\begin{array}{ccc} 1 & 0 & 0\\ 0 & 0 & 1\\ 0 & -1 & 0 \end{array}\right]$$

Similar to the previous experiments, the inverse kinematics is based on Equation (30) to obtain the joint angle, **Θ**. The change of the new weight value, $\Delta {\hat{w}}_{i}$, is obtained according to Equation (27) and used to adjust the EE motions according to Equation (28). The learning rate, i.e., η, $\eta_{1}$, $\eta_{n,1}$, …, $\eta_{n,k}$, is used to calculate the step size of manipulating the rudder in Cartesian space. Since the shape of the rudder is symmetrical at the center, the tracked locations are also symmetrical, and the EE attitude changes are the same. In Figure 16c,d, the two arms begin to manipulate the rudder at 6.75 s. In Figure 16e–h, the EE pose errors can converge to 0, and the corresponding EE motions own almost the same states ultimately by using the proposed planning method with the learning of the self-organizing competitive neural network.

Considering (1)–(3) in Section 5.2 comprehensively, the pose speed and pose error converge to 0 for the successful execution of coordination manipulation and correspond to Equation (2), and no arm motion is set as the reference for the other arms. Hence, the proposed planning method based on the self-organizing competitive neural network owns the feasibility, synchronism, and effectiveness in achieving the collaboration manipulations for the arms with physical coupling and has the contributions to the practical application.

## 6. Conclusions

This paper presents a real-time kinematically synchronous planning method for collaborative manipulation through the self-organizing competitive neural network. This method considers a type of collaborative manipulation known as the synchronization of EE motion. The sub-bases are defined for the configuration of multi-arms to obtain the Jacobian matrix of common DoFs and ensure the pose errors converging along the reducing direction of the EE total pose errors. The simulations of multi-arms with common DoFs display the consistency before the pose errors converge completely and make contributions to the collaborative manipulation of multi-arms. On this basis, an unsupervised competitive neural network is raised to regard the EE synchronous motion as the competition of neurons and adaptively increase the convergence ratio of multi-arms through the mutual learning and competition of neurons by using the inner star rules. The stability of multi-arms system is analyzed through the Lyapunov theory. Various simulations and experiments confirm that the proposed synchronous planning method is feasible, synchronous, and has the application potentiality in different cooperative manipulation tasks.

## Figures and Tables

**Figure 1 sensors-23-05120-f001:**
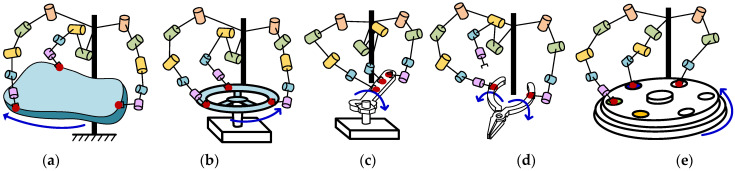
A type of cooperative manipulation. (**a**) Carrying. (**b**) Operating rudder. (**c**) Operating a wrench. (**d**) Using pliers. (**e**) Multi-station operation.

**Figure 2 sensors-23-05120-f002:**
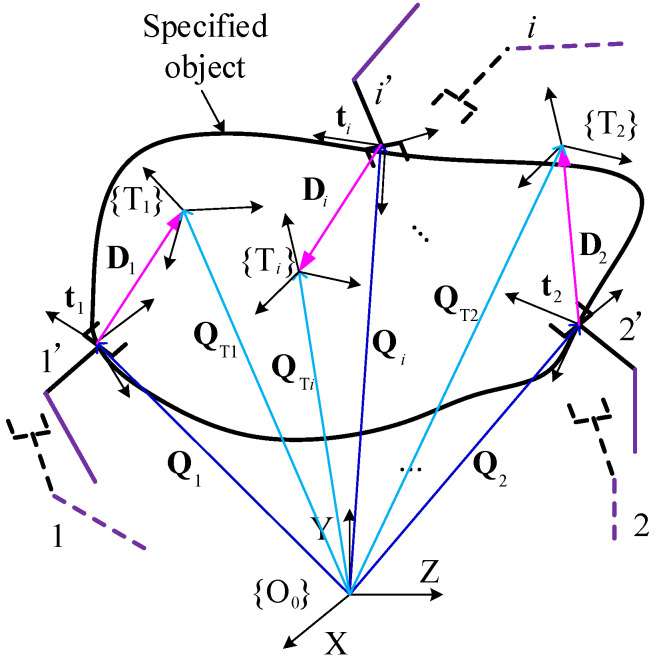
The diagram for the common features in the cooperative manipulation of multi-arms.

**Figure 3 sensors-23-05120-f003:**
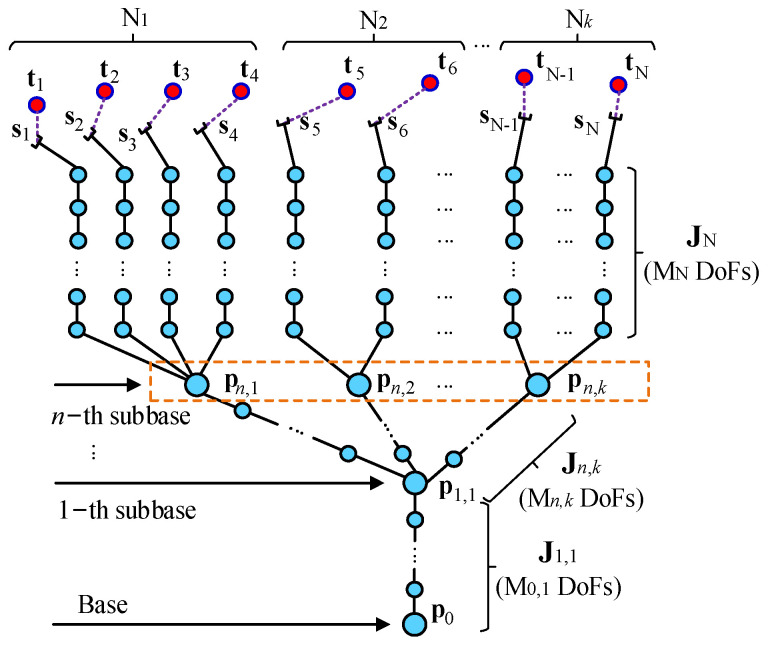
Simple configuration of multi-arm robot.

**Figure 4 sensors-23-05120-f004:**
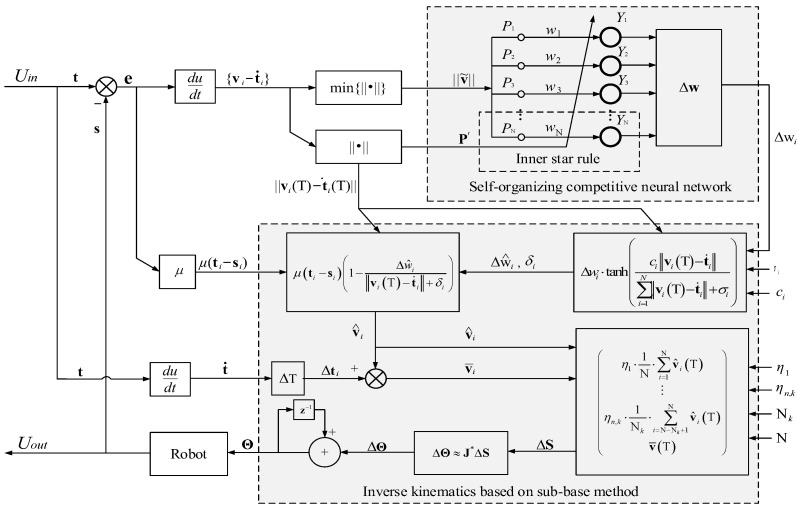
Kinematically synchronous planning for multi-arm robot. *U_in_* = **t** = (**t**_1_, **t**_2_, …, **t**_N_)^T^. *U_out_* = **s** = (**s**_1_, **s**_2_, …, **s**_N_)^T^.

**Figure 5 sensors-23-05120-f005:**
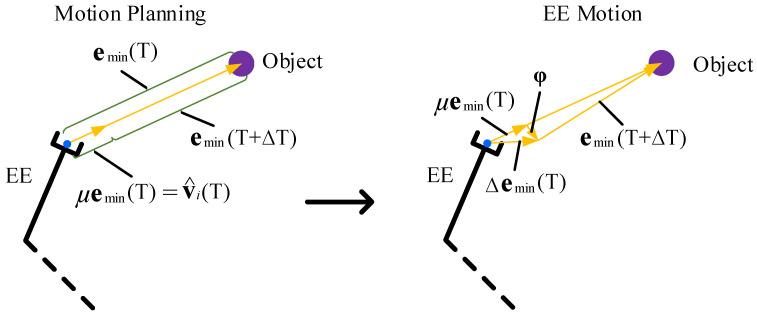
Motion planning and EE motion for the EE with the minimum pose error, $e_{\min}\left(T\right)$. ${\hat{v}}_{i}\left(T\right)=\mu e_{\min}\left(T\right)$.

**Figure 6 sensors-23-05120-f006:**
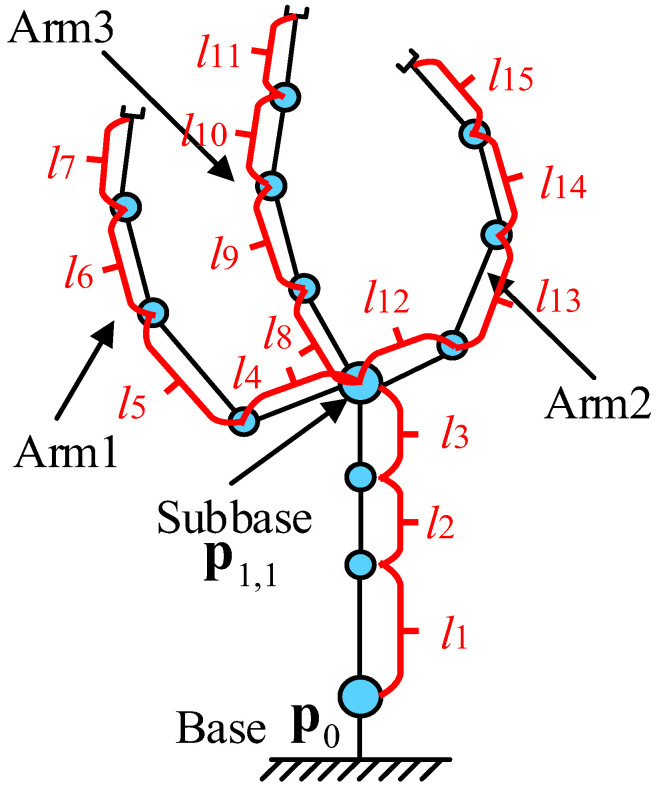
The configuration of three-arm robot with 15-DoFs.

**Figure 7 sensors-23-05120-f007:**
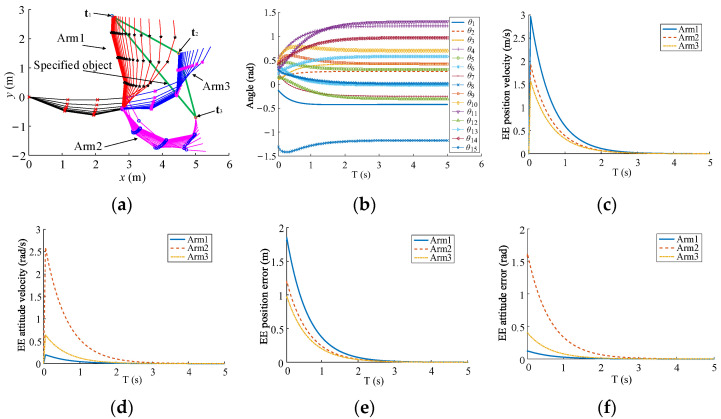
Inverse kinematics based on the traditional method in real time. (**a**) Motion of multi-arms. (**b**) Joint angles. (**c**) EE position velocity. (**d**) EE attitude velocity. (**e**) EE position error. (**f**) EE attitude error.

**Figure 8 sensors-23-05120-f008:**
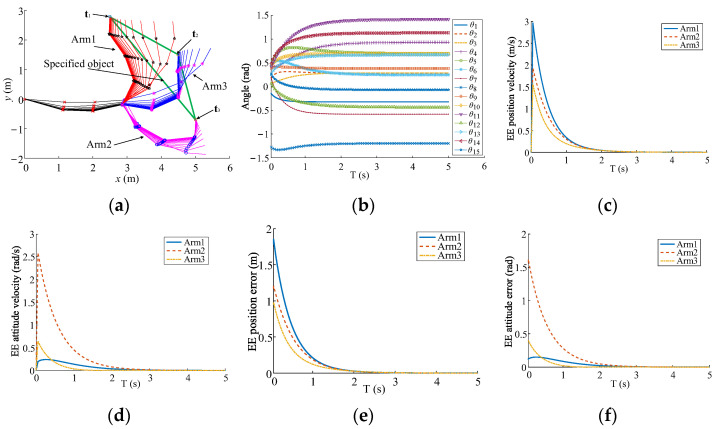
Inverse kinematics based on the sub-base method in real time. (**a**) Motion of multi-arms. (**b**) Joint angles. (**c**) EE position velocity. (**d**) EE attitude velocity. (**e**) EE position error. (**f**) EE attitude error.

**Figure 9 sensors-23-05120-f009:**
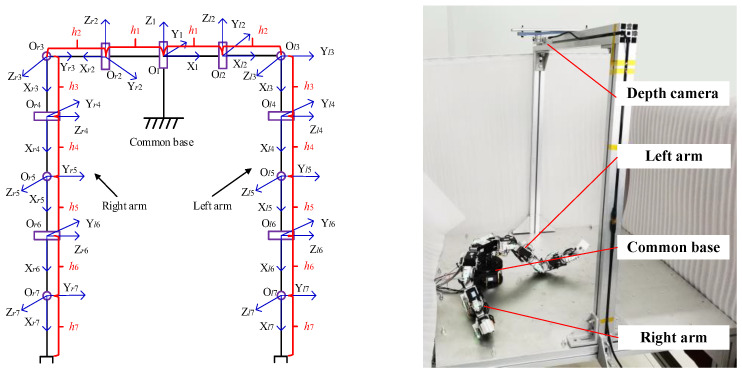
The configuration of two-arm robot with 13-DoFs.

**Figure 10 sensors-23-05120-f010:**
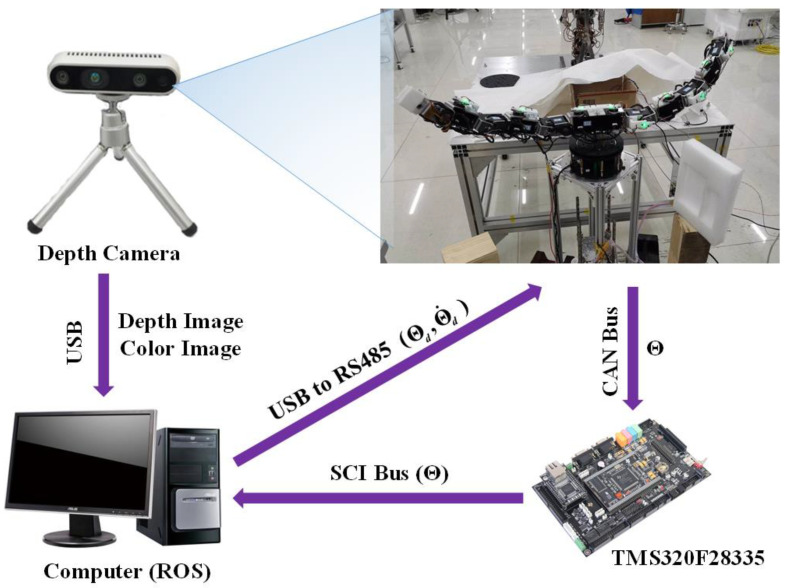
The principle of two-arm robot with 13-DoFs.

**Figure 11 sensors-23-05120-f011:**
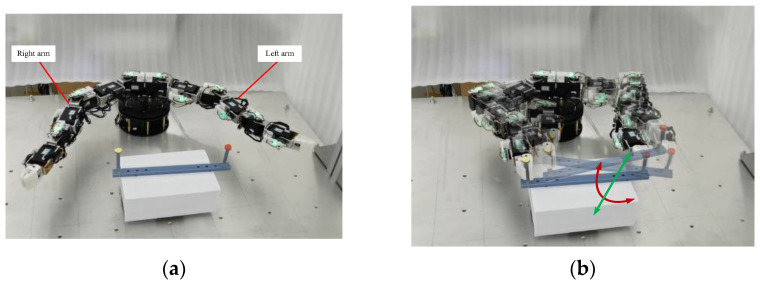
Carrying task. (**a**) Initial configuration. (**b**) Manipulating process.

**Figure 12 sensors-23-05120-f012:**
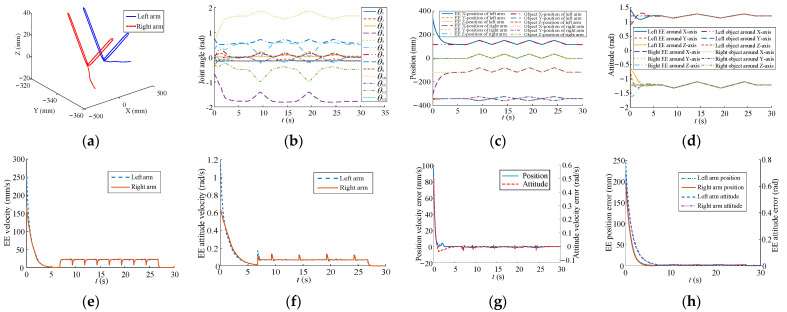
Trajectories for dual arms in carrying task. (**a**) EE movement. (**b**) Joint trajectory. (**c**) EE position. (**d**) EE attitude. (**e**) Position velocity. (**f**) Attitude velocity. (**g**) Pose velocity error. (**h**) EE pose error.

**Figure 13 sensors-23-05120-f013:**
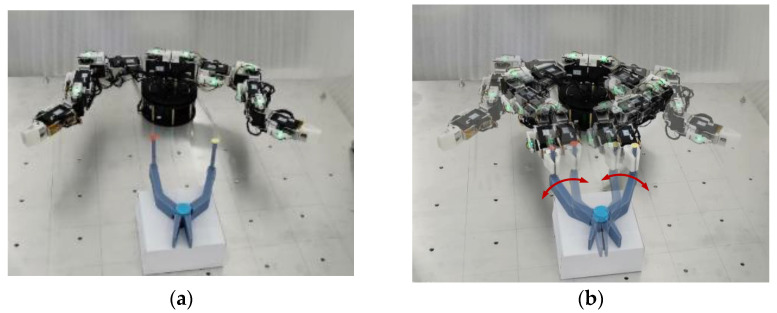
Manipulating pilers. (**a**) Initial configuration. (**b**) Manipulating process.

**Figure 14 sensors-23-05120-f014:**
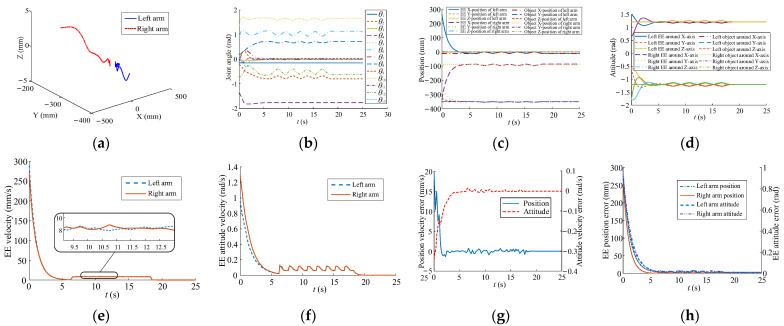
Trajectories for dual arms in manipulating pilers. (**a**) EE movement. (**b**) Joint trajectory. (**c**) EE position. (**d**) EE attitude. (**e**) Position velocity. (**f**) Attitude velocity. (**g**) Pose velocity error. (**h**) EE pose error.

**Figure 15 sensors-23-05120-f015:**
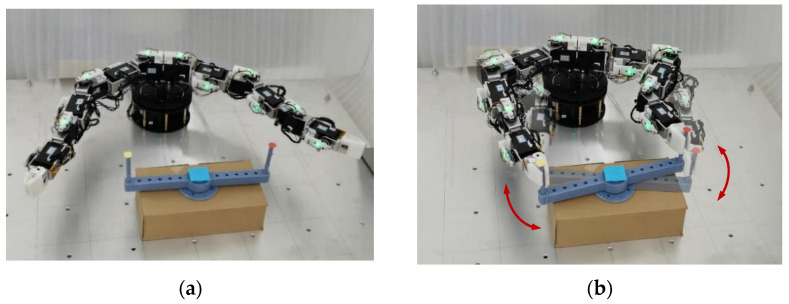
Manipulating rudder. (**a**) Initial configuration. (**b**) Manipulating process.

**Figure 16 sensors-23-05120-f016:**
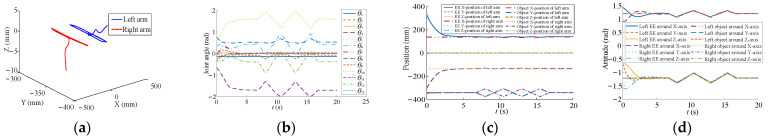
Trajectories for dual arms in manipulating rudder. (**a**) EE movement. (**b**) Joint trajectory. (**c**) EE position. (**d**) EE attitude. (**e**) Position velocity. (**f**) Attitude velocity. (**g**) Pose velocity error. (**h**) EE pose error.

**Table 1 sensors-23-05120-t001:** Parameters of three-arm robot.

*l* _i_	1	2	3	4	5
Length (m)	1.18	0.88	0.88	0.88	0.88
*l* _i_	6	7	8	9	10
Length (m)	0.88	57.85	0.88	0.88	0.88
*l* _i_	11	12	13	14	15
Length (m)	57.85	0.88	0.88	0.88	57.85

**Table 2 sensors-23-05120-t002:** Initial joint parameters of three-arm robot.

$\theta_{i}$	1	2	3	4	5
Initial angle (°)	−5.0	5.0	5.0	30.0	20.0
$\theta_{i}$	6	7	8	9	10
Initial angle (°)	20.0	20.0	−70.0	20.0	20.0
$\theta_{i}$	11	12	13	14	15
Initial angle (°)	20.0	10.0	10.0	20.0	20.0

**Table 3 sensors-23-05120-t003:** Kinematic parameters.

Parameters	*μ*	$\eta_{1,1}$	N	*b*	k	*R*
Value	0.08	1/6	3	3	1	15
Parameters	M_0,1_	M_1_	M_2_	M_3_	*λ*	ΔT
Value	3	4	4	4	0.01	0.05

**Table 4 sensors-23-05120-t004:** D-H parameters of two-arm robot *.

	*i*	$d_{i}$	$a_{i}$ (mm)	$\alpha_{i}$ (rad)	$\beta_{i}$		*i*	$d_{i}$	$a_{i}$	$\alpha_{i}$ (mm)	$\beta_{i}$ (rad)
Left Arm	1	0	$h_{1}$ = 42	0	0	Right Arm	1	0	$h_{1}$ = 42	π	0
	2	0	*h*_2_ = 84	π/2	0		2	0	$h_{2}$ = 84	π/2	0
	3	0	$h_{3}$ = 84	−π/2	0		3	0	$h_{3}$ = 84	−π/2	0
	4	0	$h_{4}$ = 84	π/2	0		4	0	$h_{4}$ = 84	π/2	0
	5	0	$h_{5}$ = 78	−π/2	0		5	0	$h_{5}$ = 78	−π/2	0
	6	0	*h*_7_ = 71	π/2	0		6	0	$h_{7}$ = 71	π/2	0
	7	0	$h_{7}$ = 71	0	0		7	0	$h_{7}$ = 71	0	0

* The base (i.e.,
$\theta_{1}$) is the common joint of two arms.

**Table 5 sensors-23-05120-t005:** Kinematic parameters of two-arm robot.

Parameters	*μ*	$\eta_{1,1}$	N	*b*	k	*r*
Value	0.005	5	2	2	1	13
Parameters	M_0,1_	M_1_	M_2_	*λ*	ΔT	—
Value	1	6	6	0.01	0.05	—

**Table 6 sensors-23-05120-t006:** The parameters of self-organizing competitive neural network.

Parameters	$\delta_{i}$	$\sigma_{i}$	$c_{i}$	η
Value	1 × 10^−5^	1 × 10^−5^	2.0	0.03

## Data Availability

Not applicable.

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
