# Peer review of "Real-Time Kinematically Synchronous Planning for Cooperative Manipulation of Multi-Arms Robot Using the Self-Organizing Competitive Neural Network"

_sensors, 2023, doi:10.3390/s23115120_

Round 1
Reviewer 1 Report
This work is really interesting; however, it is written in poor English and thus, there is difficulty in understanding some points.
I suggest that the authors should check the manuscript once again and proceed in corrections that will help the readers understand the work presented.
Reviewer 2 Report
The paper describes how the authors used an unsupervised competitive neural network model to increase the convergence ratio of multi-arms via online learning adaptively. The paper provides a theoretical analysis of the stability of multi-arms systems using Lyapunov theory and reports on various simulations and experiments that demonstrate the feasibility and applicability of the proposed synchronous planning method for different symmetric and asymmetric cooperative manipulation tasks.
Overall, the paper appears to be well-written, and the proposed method and results are well-explained. The paper's use of theoretical analysis, simulations, and experiments demonstrates the robustness and effectiveness of the proposed synchronous planning method for collaborative manipulation of multi-arms robot.
I would believe that the paper can be more elaborated in its presentation and describing its novelty and contribution to the current technology, particularly in the state-of-the-art manipulators, but it is acceptable for publication in its present form, meeting the essential requirements under the Sensors' Journal's scope.
Reviewer 3 Report
This paper presents a real-time kinematically synchronous planning method for collaborative manipulation of multi-arms robot with physical coupling using the self-organizing competitive neural network. This method defines the sub-bases for the configuration of multi-arms to obtain the Jacobian matrix of common degrees of freedom so that the sub-base motion converges along the direction for the total pose error of the end-effectors (EE). There is still some work that needs to be polished. Here are the details of my suggestions:
1. The description of innovation is not enough. It is suggested that the author explain where this method differs from previous studies about. Please explain the innovation and breakthrough of this method in the abstract or introduction explicitly. It is suggested that the author introduce some excellent latest research results, such as: Lookback option pricing models based on the uncertain fractional-order differential equation with Caputo type.
2. The application of this article is too simple to give the readers an intuitive explanation of above-mentioned analysis. It is suggested that this part be supplemented so that readers can have a clearer understanding of the theoretical part.
3. Some of the mathematical formulas seem not fit in the format. Some of them are center aligned, and some of them seem not. Mathematical formulas are necessities of the manuscript for our journal, a uniform format is more acceptable. What is recommended is that using Mathtype to write the mathematical formulas.
language expressions need to be polished. Please check over the whole manuscript and polish the language. Less errors on these will make the paper more readable.
Reviewer 4 Report
This paper presents a real-time kinematically synchronous planning method for collaborative manipulation of multi-arms robot with physical coupling using the self-organizing competitive neural network. The proposed method defines the sub-bases for the configuration of multi-arms to ensure the uniformity of the EE motion before the error converges completely and raises an unsupervised competitive neural network model to achieve the synchronous movement of multi-arms robot rapidly for collaborative manipulation. Various simulations and experiments show that the proposed method is feasible.
The paper is well structured and results seems solid, but there are some minor revisions that should be addressed to make it more understandable and clear.
1. Compared to existing research, the significant contribution of this article should be further clarified. Which aspects of the proposed method are superior to existing methods?
2. The main problem is that authors introduce their own new notations, while true scientific research starts from the investigation of what is a state of the art in the patents, books, and scientific publications.
3. In line 149, the reason of “The motion states of EEs from “1, 2, …, i ” to “ 1’, 2’, …, i’ ” are almost consistent and synchronous.” need to be explained.
4. In line 181, the parameter “r” need to be explained.
5. Do the parameters in Figure 4 need to be trained and adjusted in advance? Please analyze if the change of these parameters will affect the response time of converges.
6. What is the control cycle of the entire algorithm during actual operation?
7. The Figures 12, 14 and 16 are small and need to be enlarged.
English is fine.
Round 2
Reviewer 3 Report
This manuscript can be accepted now.
This manuscript can be accepted now.